# Anti-Caries Effect of a Mouthwash Containing *Sambucus williamsii var. coreana* Extract: A Randomized, Double-Blind, Placebo-Controlled Clinical Trial

**DOI:** 10.3390/antibiotics11040488

**Published:** 2022-04-05

**Authors:** Yu-Rin Kim, Seoul-Hee Nam

**Affiliations:** 1Department of Dental Hygiene, Silla University, Busan 46958, Korea; dbfls1712@hanmail.net; 2Department of Dental Hygiene, College of Health Sciences, Kangwon National University, Samcheok 25949, Korea

**Keywords:** *Sambucus williamsii var. coreana* extract, mouthwash, anti-caries effect, oral bacteria, preventive effect

## Abstract

This study was performed to verify the clinical effect of a mouthwash containing *Sambucus williamsii var. coreana* extract on the acid production of oral bacteria and bacteria involved in dental caries. A randomized, double-blind, and placebo-controlled trial was conducted on 66 patients of the following groups: a *Sambucus williamsii var. coreana* extract gargle group (*n* = 34) and a saline solution gargle group (*n* = 32). According to the application time of a mouthwash (before gargle application, immediately after gargle application, and five days after gargle application), we examined the emergence of dental caries-causing bacteria by PCR analysis and changes in the amount of acid production in dental plaque through a Cariview test. As a result of verifying the effect of inhibiting and preventing dental caries in the oral environment, the dental caries-causing bacteria decreased in the *Sambucus williamsii var. coreana* extract gargle group compared to the saline gargle group. In particular, *Streptococcus mutans* showed a marked decrease from immediately after application of gargle to 5 days after application. In addition, the mouthwash containing *Sambucus williamsii var. coreana* extract did not cause acid production and had low dental caries activity. A mouthwash containing *Sambucus williamsii var. coreana* extract, a natural substance, can be used as an anti-dental caries agent and be commercialized as an effective dental caries prevention agent that is safe for teeth and has an excellent antibacterial effect.

## 1. Introduction

According to the Korea Institute for Health Promotion and Development, Korea’s decayed–missing–filled teeth is improving. However, the average of Korea (1.8) is still higher than the OECD average (1.2), and the decrease in dental caries experience is lower than expected [1]. Since dental caries cannot be regenerated once afflicted, and sequelae remain, lifelong oral health depends on oral care [2].

The cause of dental caries occurs when the bacterial composition of the oral biofilm changes [3]. Bacteria involved in biofilm formation usually differ depending on the formation stage, but Gram-positive bacteria are initially attached to the tooth surface, and then Gram-positive bacteria aggregate with Gram-negative bacteria to finally complete the biofilm [4]. Among them, the representative bacteria that cause dental caries are *Streptococcus mutans* (*S. mutans*) and *Streptococcus sobrinus* (*S. sobrinus*) [5]. In addition, *Actinomyces viscosus* (*A. viscosus*) has been reported to cause root caries mainly localized in the plaque area to which the tooth is attached, especially in the shallow and middle pocket zones [6]. Hydrolyzed fructose produces lactic acid by *S. mutans*. As a result, the pH is lowered, causing tooth demineralization, which leads to dental caries, commonly called tooth decay [7]. The most basic method to remove these bacteria is a physical method such as brushing, but it is difficult to remove all the bacteria only by brushing. Therefore, mouthwash, an oral care product, can be expected to have additional effects [8].

Bacteria present in the oral cavity form a biofilm matrix by binding to the surface of the human gastrointestinal tract, laryngopharynx, and oral cavity, preventing the penetration of antibiotics and increasing drug resistance [9]. In addition, since the metabolic activity of cells in the biofilm is reduced, bacteria in the biofilm may exhibit resistance to the host defense mechanisms and antibiotics [10]. As many studies verified that chemical mouthwashes cause various problems such as brown discoloration, erosion of mucous membranes, and taste, the need for mouthwashes made of natural ingredients rather than chemical ones is increasing [11]. Therefore, mouthwashes containing natural substances that have antibacterial effects against oral disease-causing bacteria are increasing [12]. The natural substances effective for oral diseases include *Erythrina variegate* [13], *Kaempferia pandurata* [14], *Curcuma xanthorrhiza* [15], and *Drosera peltata* Smith [16].

Williams elder (*Sambucus williamsii var. coreana*) has been known to be effective for bones since ancient times. It is a deciduous broad-leaved shrub in the *Sambucus* genus in the *Caprifoliaceae* family. It grows in wetlands and valleys in mountainous areas [17]. There are several types of the *Sambucus* genus native to Korea, such as Williams elder (*Sambucus williamsii var. coreana*), Korean red elder (*Sambucus latipinna NAKAI*), and Siebold’s red elder (*Sambucus sieboldiana BLUME*). Among these *Sambucus* genus, *Sambucus williamsii var. coreana* has been used as herbal medicine. *Sambucus williamsii var. coreana* has been widely used as a home remedy known to be effective for the following symptoms: arthritis caused by wind and humidity, bleeding from external injury, postpartum blood clot, swelling and ache due to bruising and bruising itself, back pain, blood circulation problem, urticaria, fractures, and skin itching [18]. However, there is rare research verifying the antibacterial effects of *Sambucus williamsii var. coreana* extract on oral bacteria, and no research exists that examines the clinical performances.

Therefore, the purpose of this study was to verify the effect of a mouthwash containing *Sambucus williamsii var. coreana* extract on bacteria that cause dental caries in the oral cavity, acid production of bacteria, and the risk of dental caries as a randomized, double-blind, placebo-controlled clinical trial. This study provides data to prove anti-dental caries and caries-preventing effects of applying a mouthwash containing *Sambucus williamsii var. coreana* extract.

## 2. Materials and Methods

### 2.1. Study Participants

The G* Power 3.1 program calculated sample sizes. Sixty-eight participants were required for an independent *t*-test with a significance level a = 0.05 bilateral test, power = 0.8, and effect size = 0.7. In this study, the initial sample size was planned to be 96, with an expected dropout rate of 40%. The actual participants in this study were 100. A high dropout rate was set considering that the participants were college students or office workers. Among the 96 subjects, after excluding 19 subjects who did not meet inclusion criteria or refused participation for 5 days, 77 subjects were assigned to the saline gargle group and the *Sambucus williamsii var. coreana* extract gargle group at random. Additionally, 11 subjects were excluded in the 5 days of the intervention phase; data for 66 subjects were analyzed as a result (Figure 1).

This study followed the guideline of the International Council for Harmonization of Technical Requirements for Pharmaceuticals for Human Use (ICH). The Silla University Institutional Review Board (1041449-202008-HR-001, Busan, Korea) had approved the human study and WHO International Clinical Trial Registry Platform (ICTRP) was registered by clinical trial registration (10 March 2022, registration number: KCT0007064). All participants were informed of the goal and procedure of the study. Participants were also informed that they would not be penalized for refusing to participate, and they could withdraw from the study at any time.

### 2.2. Extraction of Plant Material

*Sambucus williamsii var. coreana* grown in Goesan, Chungcheongbuk-do, South Korea was purchased from Cheongmyeong Co., Ltd. (Chungju, Korea). After adding 80% ethanol to 100 g of crushed *Sambucus williamsii var. coreana*, the extraction was done in a heating mantle at 60 °C for 12 h. *Sambucus williamsii var. coreana* extract was concentrated and lyophilized using a rotary vacuum evaporator (N-1300E.V.S. EYELA Co., Tokyo Rikakikai Co., Ltd., Tokyo, Japan) after filtration using filter paper (Advantec No. 2, Tokyo, Japan). The concentrated *Sambucus williamsii var. coreana* was lyophilized using a freeze dryer (Ilshin Lab Co., Yangju-si, Korea) to obtain the *Sambucus williamsii var. coreana* powder. The final concentration was used as a mouthwash containing 10 mg/mL *Sambucus williamsii var. coreana* extract.

### 2.3. Study Design and Treatments

From October 2020 to June 2021, a dental hygienist with more than ten years of experience selected participants who visited M Dental Clinic in Busan based on those with 16 or more remaining teeth and directly explained the purpose of the study to them. Among the selected people, those with the following problems were excluded from the study: severe dental diseases (e.g., periodontitis, dental caries, dry mouth), tongue problems (e.g., tongue cancer, glossitis), taking antibiotics, having taken scaling within two months, smocking, sinus infection, and rhinitis. People with enamel caries were eligible to participate in the study, but those with more than one dentin caries were excluded. After the exclusion process, those who agreed to the study participation questionnaire became subjects while the participants who did not perform smoothly dropped out. As a result, 66 people conducted a randomized, double-blind, placebo-controlled trial.

All subjects received the same toothbrush and toothpaste used during the study period. The mouthwashes were labeled and distributed to the participants so that it was unknown whether they were in the experimental group or the control group. The subjects received instruction to gargle for 30 s before going to sleep and refrained from consuming water or food after gargling. They came to the hospital without any oral hygiene practices such as brushing teeth and gargling on the morning of the visit to the dentist for clinical examination. Data were measured by dividing them into three groups: before the application of mouthwash (‘Baseline’); immediately after the application of mouthwash (‘Treatment’); five days after the application of mouthwash as (After 5 Days). Two dental hygienists trained under the guidance of a dentist measured all clinical indicators to increase reliability.

### 2.4. Clinical Examination

To secure the homogeneity of the oral conditions of the study subjects, one week before the start of gargle application, all subjects visited M Dental Clinic in Busan and received oral examination and scaling by a dentist. After the scaling, there was a recovery period for the regeneration of the gums for one week. One week after scaling was set as ‘Baseline’. As representative teeth for an oral examination, the maxillary right first molar (#16), maxillary left central incisor (#21), mandibular left first molar (#36), and mandibular right central incisor (#41) were selected. The experimental group received 15 mL of mouthwash containing *Sambucus williamsii var. coreana* extract, the control group received 15 mL of saline, and the experiment lasted for five days.

### 2.5. Cariview Test

To examine the acid-producing capacity of the dental plaque, we used the Cariview™ kit (AIOBIO, Seoul, Korea) following the manufacturer’s instructions. After rubbing the buccal surfaces of the maxillary right first molar (#16), and mandibular left first molar (#36) with a sterile cotton swab thoroughly, the cotton swab was placed in the culture solution instantly. The culturing lasted for 48 h at 37 °C in an incubator; then, we added 10 drops of the indicator from the kit and observed how the color changed. Following the criteria presented by the manufacturer, the images taken by an optical analyzer (Allinone Bio, Seoul, Korea) were scored. The closer a Cariview score was to 0, the higher the pH value was, and a score closer to 100 meant a lower pH value. A subject was at low risk if they had a Cariview score of 0.0–40.0; medium risk exists if the score was 41.0–70.0; and the risk was high if the score was between 71.0–100.0.

### 2.6. Microbiological Analysis

For 10 s, #15 paper points were put in the gingival sulcus of four sites of two maxillary teeth (anterior and posterior tooth) and two mandibular teeth (anterior and posterior tooth) of subjects. The collected paper points were placed in sterilized tubes and stored instantly at −20 °C until just before analysis. DNA was extracted from collected #15 paper points using the AccuPrep Universal RNA Extraction Kit (Bioneer, Daejeon, Korea) following the manufacturer’s instructions. OligoMix (YD Global Life Science Co., Ltd., Seongnam, Korea) and three types of oligonucleotides (Table 1) were used, which react particularly to each bacterium [19]. We combined 9 µL of OligoMix, 10 µL of 2× probe qPCR mix (Takara Bio Inc., Shiga, Japan), and 1 µL of template DNA to prepare a sample of polymerase chain reaction (PCR) reaction. The conditions of the PCR were as follows: initial denaturation at 95 °C for 30 s, denaturation at 95 °C for 10 s, and annealing for 30 s at 62 °C. The above cycles were repeated 40 times. The Bio-Rad CFX Manager Software (Bio-Rad CFX Manager 3.1, Bio-Rad Laboratories, Hercules, CA, USA) program was used to calculate the cycle threshold (Ct) parameter, and the number of copies was derived by plotting the Ct value in the standard curve of each bacterium.

### 2.7. Statistical Analysis

The statistical analysis was performed with IBM SPSS software (IBM SPSS Statistics 24.0, SPSS Inc., Chicago, IL, USA) to evaluate significant differences. A frequency analysis was applied to the demographic characteristics of the saline gargle group and *Sambucus williamsii var. coreana* extract gargle group. We performed an independent *t*-test at a significance level of 5% and a one-way ANOVA at a significance level of 5% in analyzing the changes over time of gargle application to compare the clinical indicators of the two groups. Then, a Tukey’s test was performed as a post hoc test.

## 3. Results

### 3.1. Characteristics of Subjects

As a result of demographic analysis, both the saline gargle group and the *Sambucus williamsii var. coreana* extract gargle group had more females. The average age of the saline gargle group was 36.13 years, and that of the *Sambucus williamsii var. coreana* extract gargle group was 38.06 years. There were more people without systemic disease in both groups, and there was no significant difference in demographic variables between the two groups (Table 2).

### 3.2. Evaluation of the Acid-Producing Ability of Bacteria

Acid production of oral bacteria for dental caries activity was evaluated with the Cariview score. There was no significant difference between the saline gargle group and the *Sambucus williamsii var. coreana* extract gargle group in ‘Baseline’, ‘Treatment’, and ‘After 5 days’. There was also no significant difference according to the time of application (*p* > 0.05). In addition, there was no significant difference in the risk assessment of dental caries not only between groups but also by time (*p* > 0.05) (Table 3).

### 3.3. Changes in Dental Caries-Causing Bacteria in Subgingival Plaque

In both groups, three types of caries-causing bacteria, *S. mutans*, *S. sobrinus*, and *A. viscosus*, were discovered. Right after the gargle application (Treatment) and after five days of application (After 5 days), the changes of *S. mutans* in subgingival plaque at the maxilla and mandibular showed a significant difference (*p* < 0.05). *S. sobrinus* showed a significant difference in the maxilla at ‘After 5 days’, and there was a significant difference between the two groups only at the ‘Treatment’ in the mandibular (*p* < 0.05). *A. viscosus* showed a significant difference between the two groups in the maxilla and mandibular 5 days after gargle application (*p* < 0.05) (Table 4). Dental caries-causing bacteria in subgingival plaque showed no significant difference according to the time of gargle application in the saline gargle group. However, in the *Sambucus williamsii var. coreana* extract gargle group, *S. mutans* and *S. sobrinus* in subgingival plaques showed a difference according to the application time in both maxilla and mandibular (*p* < 0.05). *A. viscosus* showed a difference only in the maxilla according to the time of gargle application. (*p* < 0.05) (Table 4).

## 4. Discussion

When a specific bacterial community proliferates among over 700 types of bacteria in the oral cavity, the following oral diseases occur out of balance: dental caries, periodontal disease, pulpitis, root apex infection, and oral and maxillofacial tissue infection [20]. Mouthwash is a widespread oral health care product to suppress and remove bacteria. It should prevent the deposition of dental plaque, inhibit inflammatory substances in the oral cavity, and last effectively as long as possible [21]. According to Kim and Nam [22], chlorhexidine gargling contributes to the prevention of dental caries as well as the inhibition of the attachment of biofilm. However, most commercially available mouthwashes consist of chemical components. They have harmful effects on the human body causing side effects of microbial substitution and resistant bacteria, so long-term use is contraindicated [23]. Therefore, many studies about safe and effective natural materials with excellent antibacterial action in the oral cavity are ongoing actively. This study is a clinical study for a mouthwash containing *Sambucus williamsii var. coreana* extract, a natural substance that can replace chemical substances, and we tried to verify the effectiveness and safety of the mouthwash.

In this study, mouthwash using *Sambucus williamsii var. coreana* extract showed effective antibacterial action against oral bacteria. *S. mutans*, a representative bacterium that causes dental caries, was significantly reduced from ‘Treatment’ to ‘After 5 days’ in the maxilla and mandibular by applying mouthwash using *Sambucus williamsii var. coreana* extract. It means that the inhibition and death of the bacterium are shown immediately after gargling. *S. mutans* is mainly observed in the initial dental plaque. This bacterium is dangerous because it causes other oral diseases by creating an acidic environment through sugar metabolism and helps various bacteria to attach [24]. Therefore, it is crucial to suppress *S. mutans*. Mouthwash containing *Sambucus williamsii var. coreana* extract will be effective not only in preventing dental caries, but other oral diseases.

*S. sobrinus*, together with *S. mutans*, is the most commonly found bacterium in the human oral cavity and is highly related to dental caries. Many studies have reported that *S. mutans* appears more frequently in caries than *S. sobrinus*. However, some studies have reported that *S. sobrinus* is associated with high caries activity [25]. In vitro studies reported that natural substances *Humulus lupulus* [26] and *Morus alba* [27] have antibacterial activity against *S. sobrinus*. *Psoralea corylifolia* [28] has antibacterial activity against both *S. mutans* and *S. sobrinus*. However, these studies did not verify their clinical effects by actual application. Therefore, this study clinically applied mouthwash containing *Sambucus williamsii var. coreana* extract to verify the inhibitory effect on *S. mutans* and *S. sobrinus*, so this study seems to be valuable for its usefulness. Mouthwash containing *Sambucus williamsii var. coreana* extract was effective against *S. sobrinus* of subgingival plaque only at ‘After 5 days’ in the maxilla and at ‘Treatment’ in the mandibular. These results seem to be because the effect of mouthwash appears immediately, but the prolonged effect was low due to saliva in the mandibular. In the maxilla, the prolonged effect of the mouthwash was high as the contact period of the mouthwash increased.

*A. viscosus* is a bacterium that forms plaque and is also related to oral diseases such as root caries, periodontitis, sepsis, and endocarditis [29]. After applying the mouthwash containing *Sambucus williamsii var. coreana* extract to *A. viscosus*, it decreased after 5 days of application in the maxilla and the mandibular. In the saline gargle group, there was no significant difference in all dental caries-causing bacteria according to the application time. However, the *Sambucus williamsii var. coreana* extract gargle group showed differences with time.

As shown above, mouthwash containing *Sambucus williamsii var. coreana* extract reduced major caries-causing bacteria. In addition, it showed no possibility of causing caries by acid verified through the dental caries activity test. It also turned out to be safe through risk analysis of dental caries activity. Thus, we verified that mouthwash containing *Sambucus williamsii var. coreana* extract inhibits the acid-producing ability and proliferation activity of bacteria involved in dental caries. As a limitation of the clinical trial, in future studies, it is necessary to expand the types of bacteria that cause dental caries. Additional studies are required to examine the antibacterial effect and reattachment of the dental plaque when applying natural mouthwash against bacteria. In addition, more studies are necessary to verify the effects of long-term use of mouthwash containing *Sambucus williamsii var. coreana* extract and secure safety for long-term use.

## 5. Conclusions

Mouthwash containing *Sambucus williamsii var. coreana* extract did not activate dental caries and effectively inhibited the growth and proliferation of three main bacteria that cause dental caries. Therefore, mouthwash using *Sambucus williamsii var. coreana* extract could be an effective oral care product as an anti-caries agent, and it could improve the oral environment of those with dental caries.

## Figures and Tables

**Figure 1 antibiotics-11-00488-f001:**
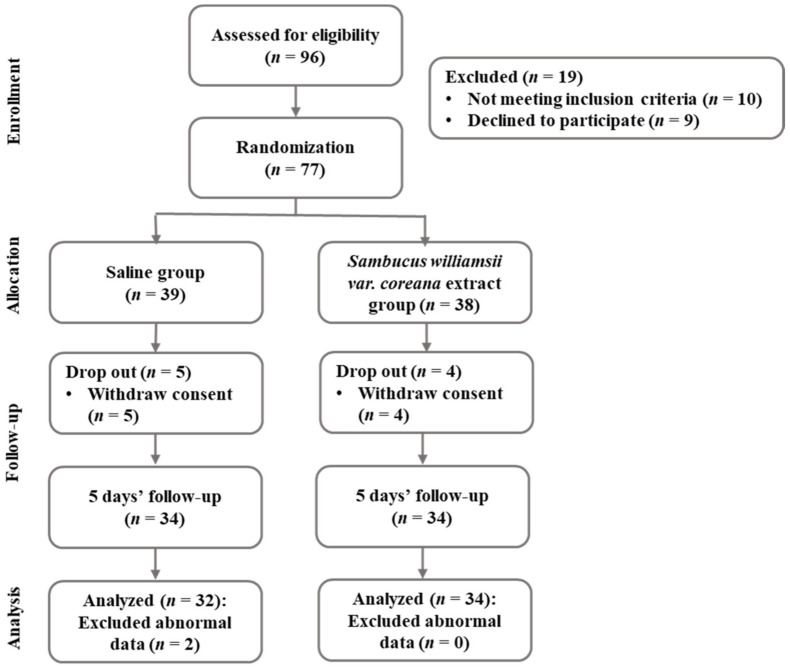
Flowchart of the research process.

**Table 1 antibiotics-11-00488-t001:** Primers and probes used in the real-time PCR assays.

Bacteria	Target Genes	Primers/Probe Sets	Amplicon Size (bp)
*Streptococcus mutans*	mannitol-specific enzyme II (mtlA) gene	5′-CAGCGCATTCAACACAAGCA-3′ 103 5′-TGTCCCATCGTTGCTGAACC-3′ 5′-HEX-TGCGGTCGTTTTTGCTCATGG-BHQ1–3′	103
*Streptococcus sobrinus*	16S ribosomal RNA gene	5′-GTACAACGAGTCGCAAGCCG-3′ 149 5′-TACAAGGCCCGGGAACGTAT-3′ 5′-FAM-TAATCGCGGATCAGCACGCC-BHQ1–3′	149
*Actinomyces viscosus*	sialidase (nanH) gene	5′-GCTCCCTCATGCTCAACTCG-3′ 5′-GATGATCTGGGCGTTGTCCA-3′ 5′-Texas Red-GAGCCGGTCCCCGACAAGAA-BHQ2–3′	140

**Table 2 antibiotics-11-00488-t002:** Characteristics of subjects.

Characteristics		N (%)		
		Saline	*Sambucus williamsii*	*p-*Value
Gender *	Male	13 (40.6)	12 (35.3)	0.800
	Female	19 (59.4)	22 (64.7)	
Age (mean ± SD) ^†^		36.13 ± 16.48	38.06 ± 17.68	0.647
Systemic disease *	No disease	29 (90.6)	31 (91.2)	1.000
	Have a disease	3 (9.4)	3 (8.8)	

^†^ *p*-values are determined by independent *t*-test, * *p*-values are determined by chi-square test (*p* < 0.05). Values are means ± standard deviations.

**Table 3 antibiotics-11-00488-t003:** Changes in the Cariview scores of two groups.

Variables	Group	Mean ± SD			* *p-*Value
		Baseline	Treatment	After 5 Days	
Cariview	Saline	54.64 ± 3.45 ^a^	54.69 ± 3.84 ^a^	52.25 ± 2.37 ^a^	0.302
	*Sambucus williamsii*	54.47 ± 8.09 ^a^	52.37 ± 4.41 ^a^	53.25 ± 6.05 ^a^	0.815
	*p-*value ^†^	0.976	0.282	0.675	
Risk	Saline	2.13 ± 0.17 ^a^	2.10 ± 0.15 ^a^	2.08 ± 0.14 ^a^	0.814
	*Sambucus williamsii*	2.10 ± 0.15 ^a^	2.00 ± 0.00 ^a^	2.13 ± 0.17 ^a^	0.145
	*p-*value ^†^	0.694	0.083	0.528	

^†^ *p*-values are determined by independent *t*-test, * *p*-values are determined by one-way ANOVA and Duncan tests (*p* < 0.05). Values are means ± standard deviations.

**Table 4 antibiotics-11-00488-t004:** Clinical outcomes of two groups.

Variables		Group	Mean ± SD			
			Baseline	Treatment	After 5 Days	* *p-*Value
*Streptococcus mutans*	Maxilla	Saline	21.13 ± 17.31 ^a^	19.97 ± 15.60 ^a^	20.77 ± 13.78 ^a^	0.989
		*Sambucus williamsii*	21.00 ± 17.75 ^a^	0.00 ± 0.00 ^b^	4.00 ± 4.80 ^b^	**0.001**
		*p-*value ^†^	0.988	**0.002**	**0.003**	
	Mandibular	Saline	10.52 ± 10.77 ^a^	9.93 ± 9.88 ^a^	10.86 ± 10.96 ^a^	0.985
		*Sambucus williamsii*	7.83 ± 6.75 ^a^	3.47 ± 3.71 ^a,b^	0.66 ± 0.76 ^b^	**0.011**
		*p-*value ^†^	0.570	**0.003**	**0.018**	
*Streptococcus sobrinus*	Maxilla	Saline	286,648.77 ± 140,717.58 ^a^	223,278.60 ± 150,159.45 ^a^	179,273.78 ± 101,632.79 ^a^	0.348
		*Sambucus williamsii*	282,228.97 ± 161,520.76 ^a^	88,299.47 ± 67,183.67 ^b^	51,774.00 ± 47,253.72 ^b^	**0.002**
		*p-*value ^†^	0.955	0.102	**0.011**	
	Mandibular	Saline	291,410.52 ± 263,791.51 ^a^	221,987.63 ± 106,938.95 ^a^	238,881.53 ± 88,927.29 ^a^	0.748
		*Sambucus williamsii*	332,049.75 ± 237,192.82 ^a^	53,006.59 ± 28,526.65 ^b^	46,060.13 ± 43,135.29 ^b^	**0.004**
		*p-*value ^†^	0.205	**0.009**	0.196	
*Actinomyces viscosus*	Maxilla	Saline	290,213.22 ± 114,182.48 ^a^	268,496.56 ± 130,468.80 ^a^	189,058.75 ± 66,533.78 ^a^	0.229
		*Sambucus williamsii*	286,789.48 ± 117,799.42 ^a^	152,394.43 ± 90,909.69 ^b^	66,951.86 ± 19,629.57 ^b^	**0.000**
		*p-*value ^†^	0.968	0.089	**0.000**	
	Mandibular	Saline	468,162.11 ± 201,367.75 ^a^	460,817.92 ± 181,326.70 ^a^	460,334.05 ± 84,132.05 ^a^	0.996
		*Sambucus williamsii*	442,364.13 ± 300,547.66 ^a^	360,205.25 ± 174,132.59 ^a^	255,872.07 ± 67,669.26 ^a^	0.485
		*p-*value ^†^	0.856	0.332	**0.001**	

^†^ *p*-values are determined by independent *t*-test, * *p*-values are determined by one-way ANOVA and Tukey tests (*p* < 0.05). Values are means ± standard deviations; significant (bold).

## Data Availability

The data presented in this study are available on request from the corresponding author.

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
