# Peer review of "Anti-Caries Effect of a Mouthwash Containing Sambucus williamsii var. coreana Extract: A Randomized, Double-Blind, Placebo-Controlled Clinical Trial"

_antibiotics, 2022, doi:10.3390/antibiotics11040488_

Round 1
Reviewer 1 Report
This paper deserves to be published. However, it would be relevant to specify wether a random drawing was done over a larger population before to constitute the final sample size after the exclusion process was effective. Is it possible to motivate ?
Author Response
We appreciate the review of the manuscript, and have been revised and changed according to the comments.

Reviewer 2 Report
Abstract: well written
Line 33: write the value from OECD and the value from Korea.
Line 70: re-write the aim. You have a RCT as a study. This should be outlined here as well.
Line 76: organise M&M chapter with subchapters that are in line with the structure of a RCT paper protocol. (you may find additional information in CONSORT protocol) (This link might help you https://www.researchgate.net/publication/282353121_A_guide_to_performing_a_peer_review_of_randomised_controlled_trials/figures?lo=1)
No need for figure 2. May be deleted.
Results chapter and Discussion should be organised as well. Inspiration from CONSORT protocol mentioned above. (there are points that are missing here from the Consort list; consult it and then improve your paper.
In the discussion chapter you should make comparisons with other mouthwashes )
Conclusion: well written.
Author Response

(The authors gave the same response as above.)

Reviewer 3 Report
The authors did an interesting study and the findings on the antibacterial properties of Sambucus williamsii var. coreana extract provide important insights to researchers in the field of caries and also periodontal diseases. However despite an apparently well-designed methodology there are some aspects that can, impact the significance of the results to their full potential.

Author Response

(The authors gave the same response as above.)
